# A COMPARISON OF SECOND-ORDER METHODS FOR DEEP CONVOLUTIONAL NEURAL NETWORKS

## ABSTRACT

Despite many second-order methods have been proposed to train neural networks, most of the results were done on smaller single layer fully connected networks, so we still cannot conclude whether it's useful in training deep convolutional networks. In this study, we conduct extensive experiments to answer the question "whether second-order method is useful for deep learning?". In our analysis, we find out although currently second-order methods are too slow to be applied in practice, it can reduce training loss in fewer number of iterations compared with SGD. In addition, we have the following interesting findings: (1) When using a large batch size, inexact-Newton methods will converge much faster than SGD. Therefore inexact-Newton method could be a better choice in distributed training of deep networks. (2) Quasi-newton methods are competitive with SGD even when using ReLu activation function (which has no curvature) on residual networks. However, current methods are too sensitive to parameters and not easy to tune for different settings. Therefore, quasi-newton methods with more self-adjusting mechanisms might be more useful than SGD in training deeper networks.

## 1 INTRODUCTION

In training deep neural networks and many other machine learning models, first-order methods have been extensively used. Stochastic Gradient Descent (SGD) method has shown positive results on various tasks in practice. It is very easy to implement and suitable to be applied in large-scale machine learning. However, SGD also has certain disadvantages. Practical success usually comes with laborious work of hyper-parameter searching. Furthermore, pure SGD often struggles to regions of loss surface with largely varying magnitudes of curvature (Dauphin et al., 2014). More importantly, it is very difficult to parallelize SGD—when increasing batch size, it is known that SGD will converge slowly and often obtain a solution with relatively poor generalization error (Keskar et al., 2016; Kawaguchi et al., 2017).

Second-order methods, on the other hand, improves the search direction using exact or approximate Hessian. Using curvature information enables such methods to make more progress per step than first-order methods relying solely on the gradient. Unfortunately, it is relatively hard to implement an efficient second-order method for large-scale deep neural networks. As parameters in the model scale up to millions, storing full hessian matrix is simply infeasible. Besides, larger datasets makes calculating exact or approximate hessian over whole dataset impossible. We will need to adopt the same mini-batch scheme as in SGD—using sub-sampled hessian in a data batch to calculate the update direction. Stochastic hessian cannot guarantee the approximated local quadratic problem to be positive semidefinite. Consequently the calculated update direction might not be a descent direction and the guarantee of the faster convergence rate of second-order method might not hold.

To overcome these challenges, few second-order methods have been proposed in the literature. Most of these methods can be categorized into into stochastic inexact-Newton methods and stochastic quasi-Newton methods, and both types of methods show certain advantages in their experimental results. However, when we try to answer the question "whether second-order method is useful for deep learning", we are still unsure the answer. The reasons are as follows.

First, the reported results in previous methods usually are based on reduce of training loss versus number of epochs. But to compute an update direction, these methods incur the extra costs of having

to compute second-order information, so overall training time might become too long to really use those methods in practice. We need a comparison of training loss versus training time to evaluate the applicability of second-order methods.

More importantly, previous methods are mostly tested on simple networks such as multi-layer fully connected networks rather than deep convolutional network, which is deemed as the standard model in applying neural networks these days. Indeed, a similar study has been conducted before (Ngiam et al., 2011), but models used in the study are mostly shallow autoencoders and only one second-order method is compared. It is still not obvious to us the advantage of second-order methods and to what extent these methods are useful in training modern deep networks, especially convolutional neural networks.

Hence, instead of proposing yet another second-order method, in this study we step back to experiment with some representative second-order methods on two datasets with various settings, and analyze both success and failure cases of second-order methods to conclude the advantages and limits of it. And we hope these information can be used in the future when designing new second-order methods.

Through this study, our major findings are as follows.

- Second-order methods can achieve better training loss in fewer number of epochs. This confirms that curvature information is still useful when training convolutional networks.
- Larger batch size benefits inexact-Newton methods more than SGD but not the case for quasi-Newton methods.
- Currently, second-order methods are too slow to be considered practical in training deep neural networks as compared to SGD.
- Network structures with zero second-order derivatives will deteriorate performance of inexact-Newton methods but not quasi-Newton methods.
- Fixed learning rate is not applicable to second-order methods

We will review the existing second-order works in the next section, and experimental and analysis will be given subsequently to support our claims above.

## 2 EXISTING SECOND-ORDER METHODS

Essentially, second-order methods obtain the update direction by minimizing a quadratic approximate function around the current solution, and the update rule can often be written as

$$w_{k+1} = w_k - \alpha_k H_k \hat{\nabla} f(w_k), \tag{1}$$

where $H_k$ is the inverse of Hessian or its approximation. For large-scale deep network training, due to the huge amount of parameters, all second-order methods need to approximate the calculation of (1) in certain ways. Based on the approximations they made, second-order methods can be categorized into following types.

### 2.1 STOCHASTIC INEXACT-NEWTON METHODS

The first family of algorithms, called "inexact-Newton method", use exact Hessian inverse as $H_k$ but compute $H_k \nabla f(w_k)$ inexactly. In (Martens & Sutskever, 2011), a "Hessian-free" approach is proposed. By using hessian-vector products, Newton updates can be solved by Conjugate Gradient (CG) inexactly, in which the search direction is computed by applying CG to the Newton method, and terminating it once it has made sufficient progress. Hessian-vector products can be computed efficiently using a form of automatic differentiation supported by most popular deep learning frameworks. (Wang et al., 2015) proposed a similar method which solves Newton equation with gradient replaced by linear combination of current and previous gradients. In practice, this method does not show prominent improvement over basic inexact-Newton method; therefore, we choose method in (Martens & Sutskever, 2011) to represent this category.

## 2.2 STOCHASTIC QUASI-NEWTON METHODS

Recently, several stochastic quasi-Newton algorithms have been developed for large-scale machine learning (Wang et al., 2017; Curtis, 2016; Keskar & Berahas, 2016; Ramamurthy & Duffy, 2016; Byrd et al., 2016). These methods use approximate hessian instead of the real one for (1), and they differ from each other by using different criteria to obtain curvature pairs and different frequencies of updating. Basically, the spirit of Quasi-Newton methods is to obtain a lower per-iteration cost but good enough second-order approximation to have better update per iteration. (Wang et al., 2017) and (Curtis, 2016) modify the BFGS update rule to prevent the updates steps from becoming unbounded values. (Byrd et al., 2016) extends L-BFGS method by decoupling of the parameter updates from the curvature estimation to achieve a better stability when dealing with large-scale problems. (Keskar & Berahas, 2016) further extended this direction by adding more checking conditions to make sure the update direction is likely descendent. In addition, they approximate empirical fisher information matrix instead of hessian matrix. (Ramamurthy & Duffy, 2016) proposed to use rank one approximation in a limited memory version manner. It combines L-SR1 method with either Line Search or Trust Region scheme to optimize the model. Notice that not all methods above are proposed specifically for non-convex problems. But even as proposed in (Keskar & Berahas, 2016) which claims its applicability for non-convex problem, in our implementation we found out the empirical fisher information becomes too small and the corresponding update is negligible. Both (Wang et al., 2017) and (Curtis, 2016) don't provide large-scale derivation in their methods and to directly implement their method, large size of hessian information needed to be stored, which is in feasible in our experiments. Therefore, we only choose (Byrd et al., 2016) and (Ramamurthy & Duffy, 2016) to represent this category.

## 2.3 STOCHASTIC GAUSS-NEWTON METHODS

Gauss-Newton is a positive semidefinite approximation to the hessian matrix. In (Botev et al., 2017), an efficient block-diagonal approximation to the Gauss-Newton matrix for multi-layer fully connected networks is proposed. It represents the approximation by Kronecker Product and factorize it to achieve an easier calculation. A similar work was done in (Martens & Grosse, 2015). The difference is that in (Martens & Grosse, 2015), Fisher-information matrix instead of hessian matrix is approximated. Although we want to include this type of optimization methods in comparison, these methods requires dedicate implementation for each type of neural network. When the network contains layers that are not fully connected and possibly with weight sharing, such as convolutional layers, it is hard to exploit the Kronecker product structure to form the Gauss-Newton matrix. Since we are focusing on convolutional neural networks in this paper, we did not take this category into comparison.

## 2.4 TRUST-REGION METHODS

Trust region methods are another important family of second-order methods. Instead of solving the linear system in (1), they obtain the update direction by minimizing the quadratic approximation around the current solution with the norm constraints. The 2-norm constraint, known as "trust region", is used to prevent the iterate from moving too far. However, this non-convex quadratic subproblem with bounded constraint is non-trivial to solve for large-scale applications, thus trust region methods have not been studied much for training deep networks. Similar to trust region method, cubic regularization method (Nesterov & Polyak, 2006) solves a similar subproblem but replaces the bounded constraints by cubic regularization. Just before submitting this work, a recent paper (Xu et al., 2017b;a) on arXiv tested a trust region method and cubic regularization method on multi-layer perceptons, but it has not been applied to convolutional neural network. We will try to include this type of methods into comparison in the future.

# 3 EXPERIMENTAL SETUPS

## 3.1 DATASETS

We will use MNIST and CIFAR-10 datasets in the experiments. MNIST consists of 60,000 1x28x28 images of hand-written digits. In this study, we report results based on all 60,000 images including

Table 1: Testing accuracy achieved by SGD for eahc model.

|         | MNIST | CIFAR-10 |
|---------|-------|----------|
| LeNet   | 99.2  | 69       |
| AlexNet | 99.4  | 77.48    |
| DRN     | 99.3  | 82.3     |

both training and validation sets. CIFAR-10 dataset consists of 60,000 3x32x32 colour images in 10 classes, with 6,000 images per class. We follow the same setting as provided with 50,000 training images and 10,000 test images. There are many data augmentation methods mentioned in the literature; however, as our analysis focuses more on optimization rather than achieving highest accuracy, we didn't augment datasets.

## 3.2 MODELS

In this study, we are interested in analyzing performance of second-order methods on convolutional neural networks. We first start with the basic LeNet5 (LeCun et al., 1995), and then test results with AlexNet (Krizhevsky et al., 2012), which can be thought as adding more layer of convolutions with larger number of filters. Last, we evaluate optimization methods on latest 18-layer Deep Residual Network(DRN) (He et al., 2016). For LeNet, we have 2 convolutional layers with 20 and 50 5x5 filters. For AlexNet, 3 convolutional layers with 64, 64, 128 5x5 filters are used. For DRN, we follow the same implementation as listed in (He et al., 2016). Best testing accuracy achieved by fixed learning rate SGD for each model is listed in Table 1.

Notice that there are certain different settings we used as compared to standard deep neural networks. First, the nonlinear activation unit used in all our implementations except few designed experiments is tanh instead of ReLu. As we will show in the following discussions, ReLu will cause the vanishing second-order information and thus prohibit the training with stochastic inexact-Newton methods. Second, we used fixed learning rate of SGD instead of having a decaying schedule. As this study mainly wants to investigate the effectiveness of second-order information over first-order gradients, we stick to the most basic usage of first-order method.

Finally, we restrict ourselves to SGD without momentum since our goal is to study whether second-order information is useful. Because of these reasons and no data augmentation, our reported best test accuracy might not be the same as in the literature. We verified that with decaying schedule, data augmentation and momentum used, our DRN implementation on CIFAR-10 can achieve around 93% test accuracy and this is almost the same as in (He et al., 2016).

## 3.3 OPTIMIZATION METHODS

We will compare SGD with one inexact-Newton method and two quasi-Newton methods. For inexact-Newton method, we compare with the hessian-free method proposed in (Martens & Sutskever, 2011), which uses Stochastic Hessian and full Gradient (SHG), and compute the update direction by conjugate gradient method. Computation of full gradient is time-consuming and sometimes is not necessary when applying this method. On MNIST with LeNet model, we show in Figure 1 that SHG with larger full gradient used is indeed better, but it's not significantly better than only partial gradients. Therefore, in our experiment when applying SHG we will only use 20% of data to compute the gradient. For quasi-Newton method, we use L-SR1 (SR1) (Ramamurthy & Duffy, 2016) and Stochastic Quasi-Newton method (SQN) (Bollapragada et al., 2016) as representative. In this study, results are based on selecting batch size as 100 except for certain designed experiments. As we are trying to understand the best applicability of each method, all the hyper-parameters are tuned to achieve the best possible results.

## 3.4 FIX LEARNING RATE VERSUS LINE SEARCH

While in (Byrd et al., 2016) a decaying schedule $1/k$ ,where k is the current number of iteration, is proposed to be the step size of each update, in practice we find out this scheme does not work on most of our experiments. Instead, we improve the stability by applying line search to find out the

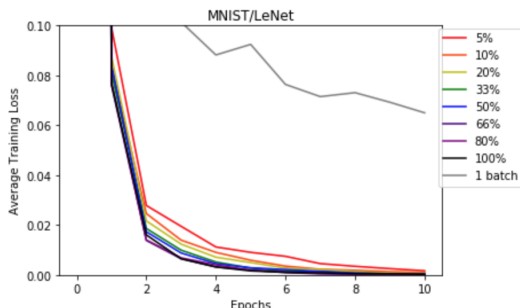

Figure 1: Percentage of data (or simply one batch) used to calculate gradient used in SHG method. We can see 20% data to compute gradient is enough. One epoch refers to iterate whole dataset once. Following figures follow the same definition.

step size. We adopt the backtracking line search and accept the step size whenever it satisfies the following sufficient decrease condition (see (Wright, 2008)):

$$f(w + \eta p) - f(w) \leq 0.1 \eta \nabla f(w)^T p, \tag{2}$$

where $p$ is the search direction, and both gradient and function values are computed using the current batch. In this study, without further notice, SGD and GD will use well tuned fix learning rate and we will apply line search for SHG, SQN and SR1. Due to this finding, we are also interested in knowing whether line search is essential when applying second-order methods. Analysis will be provided below.

## 4   RESULTS AND ANALYSIS

### 4.1   SHG OUTPERFORMS SGD IN NUMBER OF TRAINING STEPS BUT SGD IS SUBSTANTIALLY FASTER

We show MNIST results in Figure 2 and CIFAR-10 results in Figure 3. Is second-order methods useful in training deep neural network? By only looking at number of epochs to minimize training objective, we can clearly answer yes. SHG method performs well on both datasets, and quasi-Newton methods have slightly worse performance on MNIST but are significantly better than SGD on CIFAR-10. Another point to notice is that we could not find a set of parameters to make SQN work with AlexNet on CIFAR-10. This signals that quasi-Netwon methods without parameter control or self-adjusting features can easily fall short in training non-convex problems.

It might be argued that as each update, SHG accesses much more data (20% of total as mentioned) than SGD (one batch), so it is self-evident that SHG can have better results. Nevertheless, we also include the Gradient Descent (GD) method to verify that amount of data accessed is not the key to performance in this case. In each iteration, GD will use average gradient of 20% of data to be the update direction. So GD has same number of updates as SGD, and access same amount of data as SHG in each update. If amount of data accessed were the key, GD should have a similar performance as SHG method. But we can clearly tell from the both figures that GD has a very different training loss curve from SHG. GD is actually worse than SGD in most cases which indicates that aggregated gradient information is not helpful in training deep neural networks. This verifies the effectiveness of second-order information.

Thus, second-order methods sound like a promising optimization method in deep learning. But when we look at the time spent to optimize as shown in Figure 4 and 5, clearly second-order methods won't be a practical choice. SHG takes hundred to thousand times more to finish the training. This is because SHG needs to computes (almost) full gradient information. As the training data grows larger, this will be a demanding computational task. We also include time required to achieve a designated accuracy for both datasets. SGD is still substantially faster than all second-order methods. Quasi-Newton methods in general take shorter time to train and achieve decent testing results. Especially SQN is the fastest among three second-order methods. It is because SQN decouples

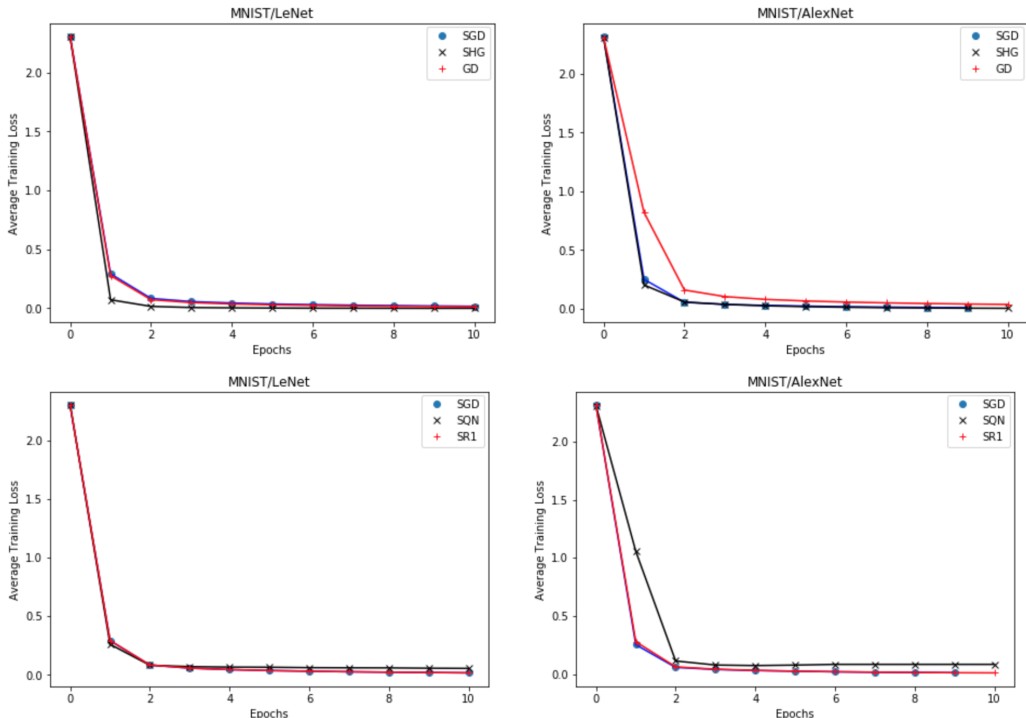

Figure 2: Average training loss versus epochs of SGD, SHG, GD, SR1, SQN on MNIST dataset. SHG is slightly faster than SGD. Quasi-Newton methods are comparable to SGD.

the computation of update pairs and parameter updates. Therefore, in each iteration the required computation is reduced.

In our implementation, computation of full gradient is done by iterating many smaller batches. Ideally, we could parallelize this operation to accelerate the SHG method so SHG might become competitive to SGD in time when distributed training is available. Furthermore, in next section we will show SHG benefits from big batches, which is also a good feature for parallelizing the computation.

|  | MNIST | | CIFAR-10 | |
| --- | --- | --- | --- | --- |
|  | LeNet | AlexNet | LeNet | AlexNet |
| SGD | 4.4 | 4.0 | 89.7 | 51.3 |
| SHG | 183 | 122 | 1422 | 1856 |
| SQN | 71.9 | 50.7 | 229 | NA |
| SR1 | 172.9 | 121.9 | 2744 | 483 |

Table 2: Time(sec) required to achieve desginated testing accuracy, MNIST:98% and CIFAR-10:73%

## 4.2 SHG PERFORMS BETTER UNDER BIG BATCH SETTING

It has been observed that increasing the batch size will hurt the convergence speed of SGD (Goyal et al., 2017; ?; You et al., 2017). Therefore, it is interesting to see the performance of second-order methods as we increase batch size.

In Figure 6, we test the performance of SQN when we increase batch size. As we can see, SQN works well when batch size is small but the training becomes stagnant when batch size becomes larger. We believe it's because there are many hyper-parameters in quasi-Newton methods such as batch size, number of curvature pairs to use, learning rate and frequency of updates. Again without a

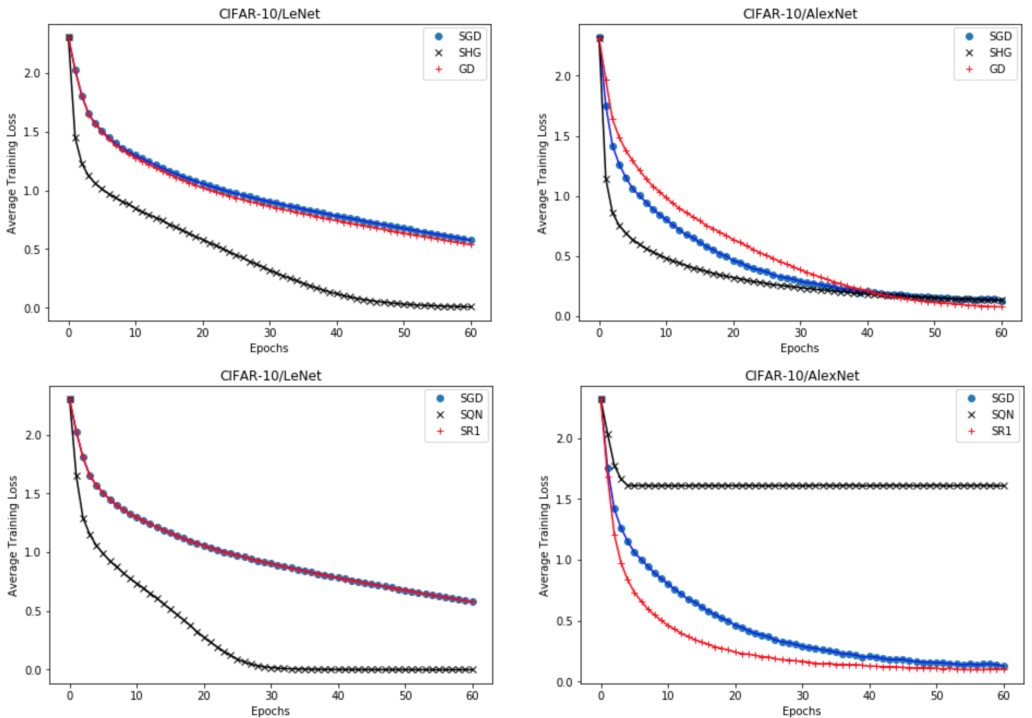

Figure 3: Average training loss versus epochs of SGD, SHG, GD, SR1, SQN on CIFAR-10 dataset. Clearly both inexact-Newton and quasi-Newton methods optimize the training loss faster than SGD.

good controlling mechanism, the approximation computed might be noisy and sudden change one of any parameters alone will make the algorithm problematic. Same situation can be observed in SR1 as shown in Figure 7. Its performance becomes worse than SGD when the batch size is increased.

On the other hand, the behavior is totally different for inexact-Newton method such as SHG—SHG benefits from bigger batches, as shown in Figure 8. Although the effect of larger batches seems to be stronger for SGD when batch size grows to 400, SHG with even larger sizes (e.g., 3200) will outperform SGD more when compared to smaller batch sizes. This implies unlike first-order methods which rely on the noisy of gradient estimation to jump out the valley, curvature information captured by SHG is enough to optimize the objective function, and larger batch size of hessian-vector product used in SHG leads to better results. When batch size is larger, it's even easier to parallelize and compute the full gradient part. This shows a good direction to apply second-order methods in the future.

### 4.3 LIMITS OF SECOND-ORDER METHODS: RELU UNIT AND IDENTITY LINK

At first glance, second-order methods especially SHG can achieve good results with standard deep convolutional neural networks. But when it comes to Deep Residual Network, the story is different as shown in Figure 9. SHG is able to train a DRN on MNIST but not CIFAR-10. As derived in (Botev et al., 2017), hessian matrix of multi-layer fully connected network will be related to this recursive pre-activation hessian H:

$$H_\lambda = B_\lambda W_{\lambda+1}^T H_{\lambda+1} W_{\lambda+1} B_\lambda + D_\lambda \qquad (3)$$

where we could define the diagonal matrices:

$$B_\lambda = \mathrm{diag}(f_\lambda'(h_\lambda)) \qquad (4)$$

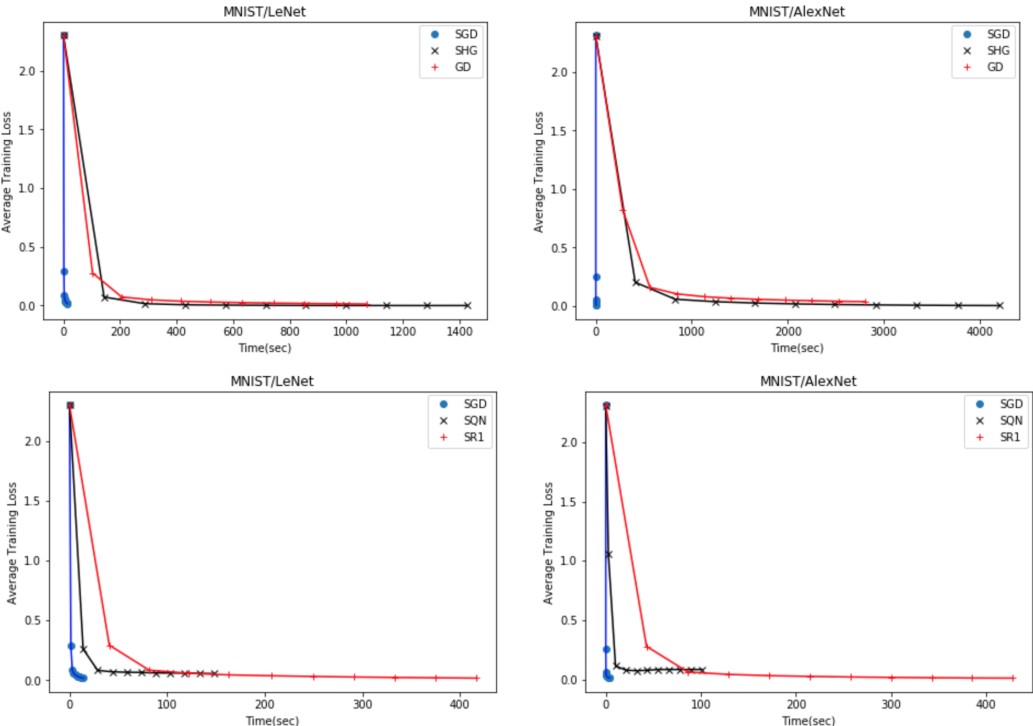

Figure 4: Average training loss versus time of SGD,SHG,GD,SR1,SQN on MNIST dataset.

$$D_\lambda = \text{diag}(f_\lambda''(h_\lambda)\frac{\partial E}{\partial a_\lambda}) \tag{5}$$

where E is the total loss, f is the activation function and $a_\lambda$ is the activation of certain layer $\lambda$. Therefore, if ReLu is used as nonlinear unit of the network, it will have zero second-order derivatives so part of the hessian information will be lost. In addition, identity link used in the DRN also has zero second-order derivatives. Since DRN contains both designs, we are not sure which unit deteriorates SHG more. Therefore, we design an ablation test to verify the effects.

First, we compare AlexNet with tanh and AlexNet with ReLu. This is a straightforward comparison to understand the influence of using ReLu. Next, we take AlexNet with tanh and substitute the last convolutional layer in AlexNet with a single residual block. Within the block, again tanh is used instead of ReLu and we keep the identity link. Although it's a relatively shallow network compared to the one used in DRN, it's enough to tell the influence of using identity link. As shown in Figure 10 and 11, both designs will deteriorate the performance of SHG. And ReLu posts stronger challenges to this type of second-order methods. This is a really problematic issue as nowadays, popularity of DRN grows fast and likely it will become the standard type of neural network. Without an efficient way of solving this problem means SHG won't be useful even it can be parallelized. On contrary, quasi-Newton methods use first-order information to approximate second-order information. Despite the second-order information is disrupted by vanishing effect, it can still receive gradients to complete the calculation. Besides, we observe that it converges to smaller training loss than SGD on both MNIST and CIFAR-10. It has the potential to be applied in training deeper residual networks.

The reason why SHG still works on training MNIST is likely due to the fact that images in MNIST are simple patterns with most parts black. So the information lost is not severe enough to stop us from training. But in general it should become an issue on most of real-world datasets.

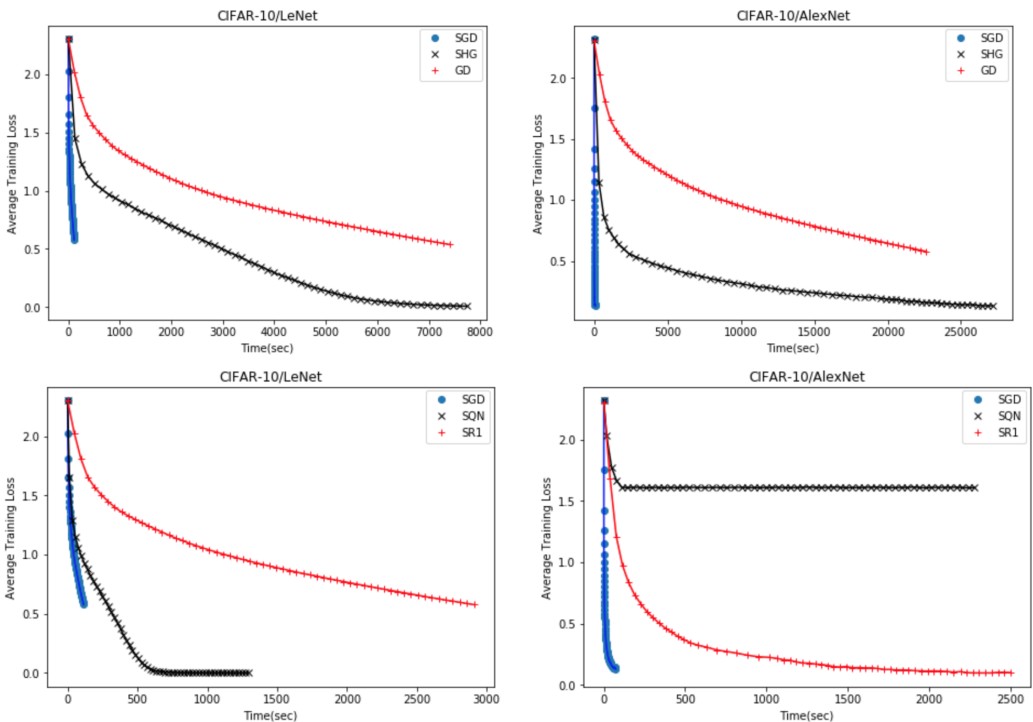

Figure 5: Average training loss versus time of SGD,SHG,GD,SR1,SQN on CIFAR-10 dataset. SGD in practice requires minimal computations and consequently faster than second-order methods.

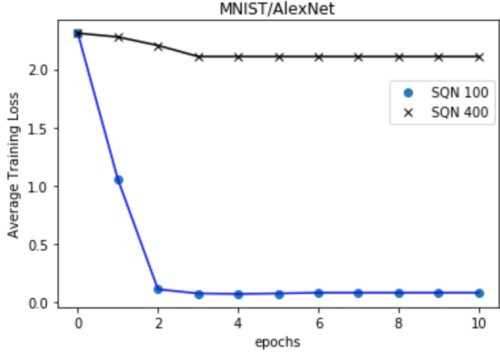

Figure 6: Effect of changing batch size of SQN method. SQN fails to generalize the performance to larger batches. In principle SQN is sentitive to hyper-parameters.

## 4.4 LINE SEARCH IS ESSENTIAL TO SECOND-ORDER METHODS

In this part, we replace the line search computations with well tuned fixed learning rate. In sum, second-order methods will not work on most of cases as shown in figure 12. As each update might not be descent direction, larger step sizes usually cause model to explode at certain point of training. With a fixed step size, only smaller values could be chosen. Consequently, the training process is relatively stagnant. The full capability of second-order methods cannot be achieved.

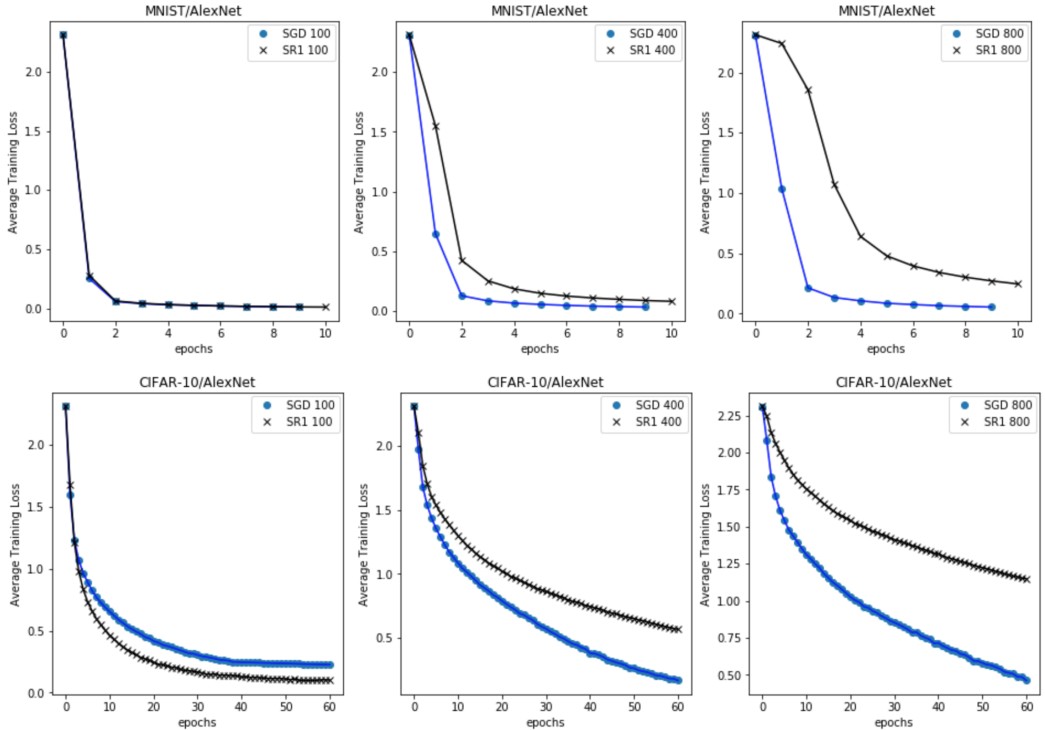

Figure 7: Effect of changing batch size of SGD and SR1 methods. SR1 is sensitive to batch size too. As batch size grows, SGD outperforms SR1 on both datasets.

## 5 CONCLUSIONS

Second-order information is helpful to reduce the number of training epochs needed but it severely suffers from computational cost. So presently, light-weight first-order methods still dominate the optimization of deep neural network. Fortunately, both inexact-Newton methods and quasi-Newton methods seem to have certain ways to improve in the future. For inexact-Newton methods, parallelization is obviously the most promising direction to further study. With bigger batches and proper distributed system supported, it has a chance to achieve competitive results in practical timing constraints. Nevertheless, we will need to figure out how to deal with vanishing second-order information to make it really useful in deep learning.

For quasi-Newton method, although currently it is fast enough to be considered as the same order as SGD, a good balance between the precision of approximation and frequency of updates should still be considered to further cut the computational time. In addition, a more efficient and safer self-adjusting or correction mechanism can be added to make it stable under different setup of parameters and batch sizes. In addition, from preliminary analysis, quasi-Newton methods show competitive results when training DRN on both MNIST and CIFAR-10. This is another interesting research direction to explore.

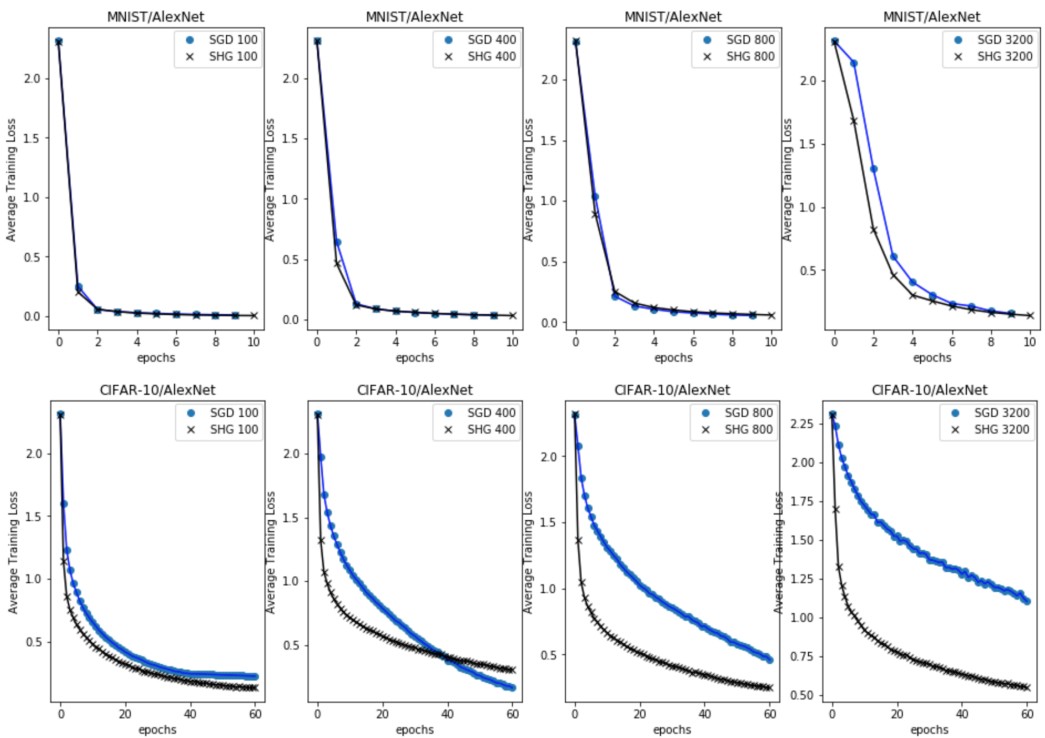

Figure 8: Effect of changing batch size of SGD and SHG methods. SGD benefits a bit when batch size grows to 400 but not larger. SHG outperforms SGD significantly when batch size keeps growing. Notice that large batch size means smaller number of updates in each epoch.

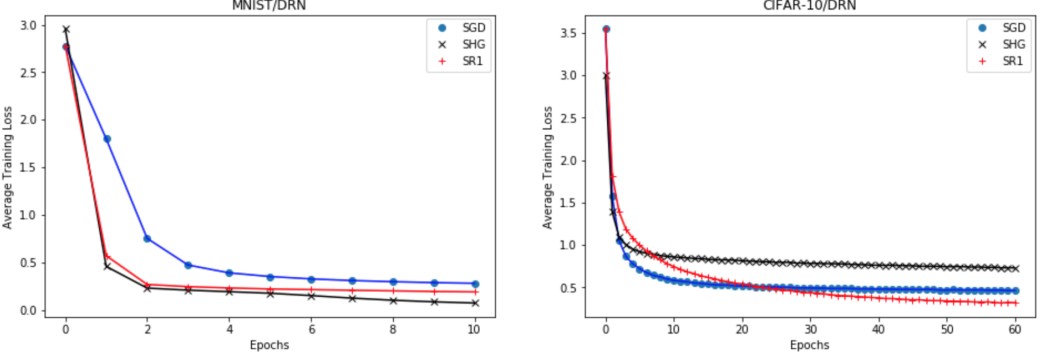

Figure 9: Results of training regular DRN with ReLu activation. Although SHG can still optimize DRN on MNIST well, it fails to converge to better training loss on CIFAR-10 dataset.

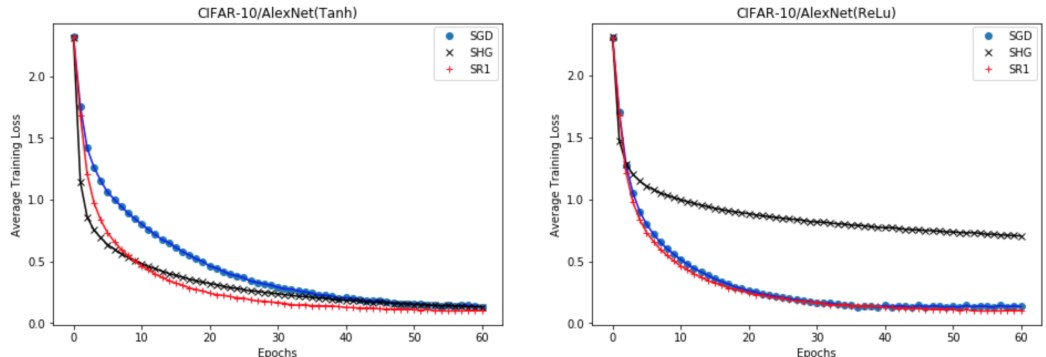

Figure 10: Effect of using ReLu in AlextNet. Left figure is Regular AlexNet with tanh activation and right figure is AlexNet with ReLu activation. Apparently using ReLu in deep neural networks deteriorates performance of SHG but not SR1.

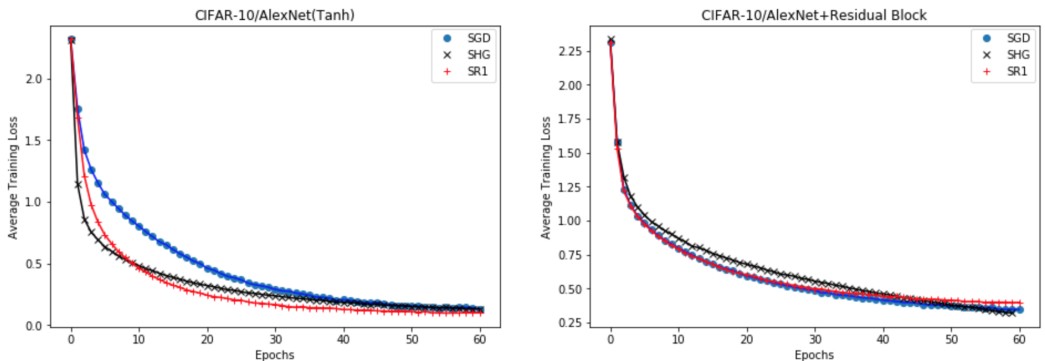

Figure 11: Effect of adding Identity expression in AlexNet. Left figure is Regular AlexNet with tanh activation and right figure is AlexNet with last convolutional layer replaced by a residual block. The influence of identity link is not as strong as ReLu but still affects the performance of SHG.

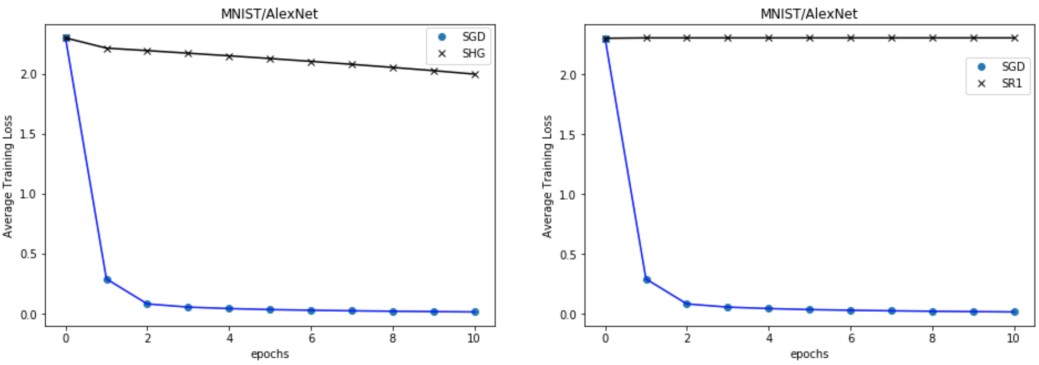

Figure 12: Effect of using fixed learning rate of SGD and SR1 methods. Second-order methods cannot work with fixed learning rate.

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
