# OpenReview forum: "A comparison of second-order methods for deep convolutional neural networks"
_ICLR.cc/2018/Conference — Reject_

### Official Review · AnonReviewer1 · 2017-11-24
**The paper attempts a well-needed experimentation of second order methods for training large CNNs. However, the results are mostly negative, it is not bold, and it is incomplete.**

**Rating:** 5
**Confidence:** 5

**Review:**

A good experimentation of second order methods for training large DNNs in comparison with the popular SGD method has been lacking in the literature. This paper tries to fill that gap. Though there are some good experiments, I feel it could have been much better and more complete.

Several candidates for second order methods are considered. However, their discussion and the final choice of the three methods is too rapid. It would have been useful to include an appendix with more details about them.

The results are mostly negative. The second order methods are much slower (in time) than SGD. The quasi-Newton methods are way too sensitive to hyperparameters. SHG is better in that sense, but it is far too slow. Distributed training is mentioned as an alternative, but that is just a casual statement - communication bottleck can still be a huge issue with large DNN models.

I wish the paper had been bolder in terms of making improvements to one or more of the second order methods in order to make them better. For example, is it possible to come up with ways of choosing hyperparameters associated with the quasi-Newton implementations so as to make them robust with respect to batch size? Second order methods are almost dismissed off for RelU - could things be better with the use of a smooth version of RelU? Also, what about non-differentiability brought in my max pooling?

One disappointing thing about the paper is the lack of any analysis of the generalization performance associated with the methods, especially with the authors being aware of the works of Keskar et al and Kawaguchi et al. Clearly, the training method is having an effect on generlaization performance, with noise associated with stochastic methods being a great player for leading solutions to flat regions where generalization is better. One obvious question I have is: could it be that, methods such as SHG which have much less noise in them, have poor generalization properties? If so, how do we correct that?

Overall, I like the attempt of exploring second order methods, but it could have come out a lot better.

---

### Official Review · AnonReviewer2 · 2017-11-24
**See below.**

**Rating:** 6
**Confidence:** 3

**Review:**

The paper conducts an empirical study on 2nd-order algorithms for deep learning, in particular on CNNs to answer the question whether 2nd-order methods are useful for deep learning.  More modestly and realistically, the authors compared stochastic Newton method (SHG) and stochastic Quasi- Newton method (SR1, SQN) with stochastic gradient method (SGD).  The activation function ReLu is known to be singular at 0, which may lead to poor curvature information, but the authors gave a good numerical comparison between the performances of 2nd-order methods with ReLu and the smooth function, Tanh.  The paper presented a reasonably good overview of existing 2nd-order methods, with clear numerical examples and reasonably well written.

The paper presents several interesting empirical findings, which will no doubt lead to follow up work. However, there are also a few critical issues that may undermine their claims, and that need to be addressed before we can really answer the original question of whether 2nd-order methods are useful for deep learning.

1. There is no complexity comparison, e.g. what is the complexity for a single step of different method.

2. Relatedly, the paper reports the performance over epochs, but it is not clear what "per epoch" means for 2nd-order methods.  In particular, it seems to me that they did not count the inner CG iterations, and it is known that this is crucial in running time and important for quality.  If so, then the comparison between 1st-order and 2nd-order methods are not fair or incomplete.

3. The results on 2nd-order methods behave similarly to 1st-order methods, which makes me wonder how many CG iterations they used for 2nd-order method in their experiment, and also the details of the data.  In particular, are they looking at parameter/hyperparameter settings for which 2nd-order methods aren't really necessary.

4. In deep learning setting, the training objective is non-convex, which means the Hessian can be non-PSD.  It is not clear how the stochastic inexact-Newton method mentioned in Section 2.1 could work.  Details on implementations of 2nd-order methods are important here.

5. For 2nd-order methods, the author used line search to tune the step size.  It is not clear in the line search, the author used the whole training objective or batch loss.  Assuming using the batch loss, I suspect the training curve will be very noisy (depending on how large the batch size is).  But the paper only show the average training curves, which might be misleading.

Here are other points.

1. There is no figure showing training/ test accuracy.  Aside from being interested in test error, it is also of interest to see how 2nd order methods are similar/different than 1st order methods on training versus test.

2. Since it is a comparison paper, it only compares three 2nd-order methods with SGD.  The choices made were reasonable, but 2nd-order methods are not as trivial to implement as SGD, and it isn't clear whether they have really "spanned the space" of second order methods

3. In the paper, the settings of LeNet, AlexNet are different with those in the original paper.  The authors did not give a reason.

4. The quality of figures is not good.

5. The setting of optimization is not clear, e.g. the learning rate of SGD, the parameter of backtrack line search.  It's hard to reproduce results when these are not described.

---

> ### Author Response · Authors · 2017-12-31
> **Specific reply to Reviewer 2**
>
>
> Q: Relatedly, the paper reports the performance over epochs, but it is not clear what "per epoch" means for 2nd-order methods.  In particular, it seems to me that they did not count the inner CG iterations, and it is known that this is crucial in running time and important for quality.
>
> A: We run each iteration 10 CG steps but in fact even 1 CG step performs roughly the same as 10 steps. Each hessian-vector product is about 2 times more expensive than a gradient computation. Somehow the key bottleneck here in SHG is on computing the full gradient (or 20% gradient in our paper) instead of CG (using much smaller subsamples).  Huge time difference basically comes from gradient aggregation step not CG or line search stage.
>
>
> Q: In the paper, the settings of LeNet, AlexNet are different with those in the original paper.  The authors did not give a reason.
>
> A: We basically follow the same architecture of LeNet and AlexNet, and the same as residual network 20-layer implementation. The only difference is that as we notice SHG cannot work on networks with ReLu unit, so we replace it with tanh. Also, we didn’t use data augmentation to accelerate the experiment.
>
> Q: The results on 2nd-order methods behave similarly to 1st-order methods, which makes me wonder how many CG iterations they used for 2nd-order method in their experiment, and also the details of the data.  In particular, are they looking at parameter/hyperparameter settings for which 2nd-order methods aren't really necessary.
>
>
> A: CG part is explained above. SHG basically doesn’t have any hyperparameter to tune as we adopt fixed CG step and line search scheme. For other methods, there might be some parameters. For example, SQN needs to decide the update frequency and length of memory. Default values provided in the original paper does not converge in deep neural networks.
>
> Q: In deep learning setting, the training objective is non-convex, which means the Hessian can be non-PSD.  It is not clear how the stochastic inexact-Newton method mentioned in Section 2.1 could work.  Details on implementations of 2nd-order methods are important here.
>
> A: Indeed, Hessian might not be PSD. That’s why line search is important for inexact-Newton to work. As we decrease the step-size, eventually it will find an update step either descendent or the step size is too small to make this update affect the performance.

---

### Official Review · AnonReviewer3 · 2017-11-27
**Important topic, not enough evidence**

**Rating:** 3
**Confidence:** 4

**Review:**

This paper presents a comparative study on second-order optimization methods for CNNs. Overall, the topic is interesting and would be useful for the community.

However, I think there are important issues about the paper:

1) The paper is not very well-written. The language is sometimes very informal, there are many grammatical mistakes and typos. The paper should be carefully proofread.

2) For such a comparative study, the number of algorithms and the number of datasets are quite little. The authors do not mention several important methods such as (not exhaustive)

Schraudolph, N. N., Yu, J., and Günter, S. A stochastic quasi-Newton method for online convex optimization.
Gurbuzbalaban et al, A globally convergent incremental Newton method (and other papers of the same authors)
A Linearly-Convergent Stochastic L-BFGS Algorithm, Moritz et al

3) The experiment details are not provided. It is not clear what parameters are used and how.

4) There are some vague statements such as "this scheme does not work" or "fixed learning rates are not applicable". For instance, for the latter I cannot see a reason and the paper does not provide any convincing results.

Even though the paper attempts to address an important point in deep learning, I do not believe that the presented results form evidence for such rather bold statements.

---

> ### Author Response · Authors · 2017-12-31
> **Specific reply to Reviewer3**
>
> Reviewer 3 mentioned that certain description of certain existing methods such as “this scheme does not work” or “fixed learning rates are not applicable” are not clear. This is the case which we just explained above that implementation is not trivial. When implementing some algorithms, we find out our implementation simply cannot converge during training, or in original paper authors use fixed learning rate but it again diverges during training. This signals the fact that most existing second-order methods are not stable under more complicated problems but this observation is hardly discussed before. This makes extensive study of 2nd-order methods infeasible.

---

### Author Response · Authors · 2017-12-31
**Reply to reviewers**

We thank all reviewers for valuable comments. We will reply to common questions first and then reply specifically to reviewer 2 and 3.

As mentioned by reviewer 2, our choice of second-order methods is reasonable and it’s based on general categories of second-order methods. However, the implementation of all these second-order methods are not trivial so it’s pretty much impossible for us to reimplement all unless authors release their code, which is unfortunately not the case for almost all cases. In addition, most methods require delicate implementation for different models, and this inhibits us from experimenting all different methods on various CNN models. So we mainly focus on the methods which claim to be useful for non-convex problems or especially for deep neural networks. Thus works mentioned by the reviewer 3 are not considered since those works focus on “strongly convex” problems. From the remaining methods, we chose exemplar methods which we could implement and validate their correctness by comparing the experimental results on their original work. This coverage might not be exhaustive, but the real situation is that even the characteristics of vanilla-version 2nd-order methods on convolutional neural network are not well understood. We believe results in our paper provide some new findings which can benefit later development of 2nd-order methods.


Next we explain more details on implementation. Reviewers raise concerns on details of our SGD setup. For SGD, we tried learning rate starting from 0.001 and upscale an order until the training curve does not converge. The value of optimal learning rate can be different with different model/dataset. Essentially we do this for every set up. We didn’t repeat this for second-order methods as in second-order methods we adopt line search scheme which does not need to predetermine a fixed learning rate.

---

### Decision · Program_Chairs · 2018-01-29
**ICLR 2018 Conference Acceptance Decision**

**Decision:**

Reject

**Comment:**

This paper investigates the performance of various second-order optimization methods for training neural networks. Comparing different optimizers is worthwhile, but as this is an empirical paper which doesn't present novel techniques, the bar is very high for the experimental methodology. Unfortunately, I don't think this paper clears the bar: as pointed out by the reviewers, the comparisons miss several important methods, and the experiments miss out on important aspects of the comparison (e.g. wall clock time, generalization). I don't think there is enough of a contribution here to merit publication at ICLR, though it could become a strong submission if the reviewers' points were adequately addressed.